# Factors Influencing Online Hotel Booking: Extending UTAUT2 with Age, Gender, and Experience as Moderators

**Chia-Ming Chang [1], Li-Wei Liu [2], Hsiu-Chin Huang [3] and Huey-Hong Hsieh [4],***

[1] Department of Physical Education, Health & Recreation, National Chiayi University, Chiayi 62103, Taiwan
[2] Department of Leisure Service Management, Chaoyang University of Technology, Taichung 41349, Taiwan
[3] Physical Education and Arts School, Chengyi University College, Jimei University, Xiamen 361021, China
[4] Department of Leisure Management, Taiwan Shoufu University, Tainan 72153, Taiwan
* Correspondence: nancylin809@gmail.com; Tel.: +886-6-571-8888 (ext. 249)

**Abstract:** As people feel more comfortable using the Internet, online hotel bookings has become popular in recent years. Understanding the drivers of online booking intention and behavior can help hotel managers to apply corresponding strategies to increase hotel booking rates. Thus the purpose of this study is to investigate the factors influencing the use intention and behavioral intention of online hotel booking. The proposed model has assimilated factors from the extended Unified theory of Acceptance and use of Technology (UTAUT2) along with age, gender, and experience as moderators. Data were collected by conducting a field survey questionnaire completed by 488 participants. The results showed that behavioral intention is significantly and positively influenced by performance expectancy, social influence, facilitating condition, hedonic motivation, price value, and habit behavior. Use behavior is positively influenced by facilitating condition and hedonic motivation. As for moderators, gender moderates the relationships between performance expectancy, social influence, and behavioral intention. Age moderates the relationships between effort expectancy, social influence, hedonic motivation, and behavioral intention. Experience moderates the relationships between social influence, price value, and behavioral intention and between habit behavior and use behavior. Based on the results, recommendations for hotel managers are proposed. Furthermore, research limitations and future directions are discussed.

**Keywords:** international hotel; customers; online hotel bookings; UTAUT2

## 1. Introduction

Advanced technology these days has made information technology more accessible to people than before. According to Market Intelligence Consulting, Taiwan [1], 45% of the daily purchases are made through Internet, and $859 is spent annually per person. Interestingly, this record is nowhere to be found in the Asia-Pacific area [2].

Since technology use is inevitable in hotel industry, hotel managers are concerned about how Internet use can bring in more customers. Review of the literature suggested that job satisfaction of employees and employers, organizational commitment, customer loyalty, and brand image have been a focus of research [3–6]. How hotel online reservation system affects consumer decision-making process when choosing a hotel has not been addressed much in prior research. It is therefore suggested that hotel business operators improve competitiveness using Internet. By understanding the relationship between Internet use and customer satisfaction, hotel industry as a whole may improve business efficiency and competitiveness.

Hence, the present study explored how hotel online reservation system affected consumer intention and behavior, and the research respondents were customers in some selected international hotels. As hotel online reservation system runs nonstop and responds to customer needs in a timely manner, it is important for hotel operators to understand if online reservation system is a dominant factor to affect potential customers' use intention and use behavior.

Prior research on the use of information technology and technology acceptance has been conducted from psychological and social perspectives. Technology acceptance model (TAM) [7] is based on the theory of reasoned action (TRA), and the model suggests that when users are presented with a new computer and information technology, the factors that influence their decision about how they use technology are perceived usefulness and perceived ease-of-use, which in turn influence user attitude and behavior toward information technology. TAM theory is considered as an effective model in a variety of research and also applies to numerous countries [8,9]. According to Venkatesh, Morris, Davis, and Davis [10], it is however insufficient to explain individual use intention and behavior toward information technology simply from psychological and social perspectives. Consequently, in 2003 they developed the unified theory of acceptance and use of technology (UTAUT). The theory holds four key constructs: (1) performance expectancy, (2) effort expectancy, (3) social influence, and (4) facilitating conditions, and aims to explain user intention to use technology. The theory also applies gender, age, experience, and voluntariness of use which are posited to moderate the impact of the four key constructs on use intention and behavior, and reports to account for 70% of the variance [11].

Venkatesh et al. [12] later found that more and more studies have been conducted using technology acceptance model, and thus extended the UTAUT in a consumer context. They proposed the unified theory of acceptance and use of technology 2 (UTAUT2). They incorporated constructs, such as hedonic motivation (HM), price value (PV), and habit (HT) to study user intention and behavior [12]. Compared with UTAUT, they found UTAUT2 provided a more satisfactory explanatory power than UTAUT.

While UTAUT has been employed and verified across multiple disciplines, UTAUT2 is not commonly seen in recent research publications. It is therefore hoped that researchers across disciplines apply UTAUT2 in their particular area of study to better understand whether the model is applicable across different research areas.

## 2. Research Model and Research Hypotheses

In this study, UTAUT2 was employed to examine online hotel booking impact factors. First, the background of UTAUT2 model is described, followed by the related constructs and hypotheses.

### 2.1. Background of UTAUT2 Model

Venkatesh, Morris, Davis, and Davis [10] integrated eight technology acceptance models and proposed the Unified Theory of Acceptance and Use of Technology (UTAUT) model. Under UTAUT model, there are four independent variables, namely, performance expectancy (PE), effort expectancy (EE), social influence (SI), and facilitating conditions (FC). The four variables can influence consumers' use intention and use behavior. The four other variables, namely, gender, age, experience, and voluntariness play the moderating role, which moderate the relationship between the independent variables and dependent variables. Venkatesh et al. [10] explained the relevant variables under UTAUT model as follows:

- PE: the degree to which using a technology will provide benefits to consumers.
- EE: the degree of ease associate with consumers' use of technology.
- SI: the extent to which consumers perceive that important others (family and friends) believe they should use a particular technology.
- FC: consumers' perceptions of the resources and support available to perform a behavior.
- According to UTAUT, PE, EE, SI, and FC are theorized to influence the behavioral intention to use a technology, while behavioral intention and FC determines technology use.

Basically, UTAUT takes an approach that emphasizes the importance of utilitarian value and was developed for an employee acceptance and use setting. Therefore, Venkatesh, Thong, Xu [12] incorporated four other variables—hedonic motivation (HM), price vale (PV), experience, and habit—with the UTAUT model and extended UTAUT into UTAUT2 model to be used for individual use of technology. They explained the variables as follows:

- HM: the fun or pleasure derived from using a technology.
- PV: consumers' cognitive tradeoff between the perceived benefits of the applications and the monetary cost for using them.
- Experience: an opportunity to use a target technology.
- Habit: the extent to which people tend to perform behaviors automatically because of learning.
- In UTAUT2, PE, EE, SI, FC HM, PV and habit influence behavioral intention, while behavioral intention, FC, PV and habit influence use behavior. Age, gender, and experience play as moderators. (See Figure 1 for more detailed relations).
- Since this study focuses on consumers' technology acceptance and use context, the UTAUT2 model is adapted as the fundamental prototype of our research framework. The constructs used in the model are explored and our hypotheses based on the model are tested to identify the key factors affecting consumers' online hotel booking use intention and behavior.
- On the basis of the UTAUT2 model, this paper proposed the hypotheses in the following sections.

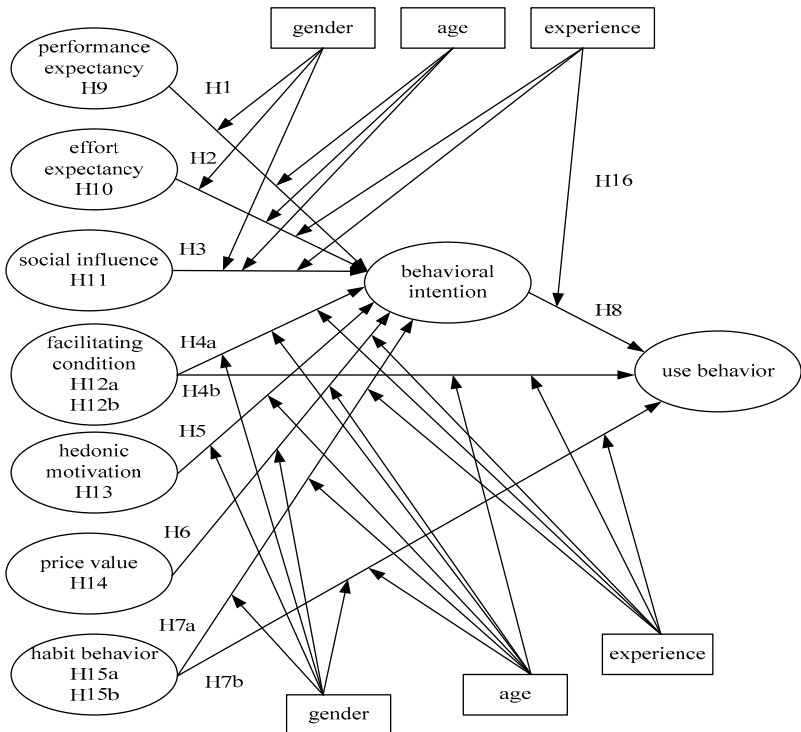

**Figure 1.** The hypothesized conceptual model of the study.

## 2.2. Performance Expectancy (PE) and Behavioral Intention (BI)

Venkatesh et al. [10] stated that when a customer is presented with a new information technology and learned using this technology, it resulted in a better performance; he is more likely to continue using this technology in the future. Moran [13] also found that performance expectancy is a key determinant affecting college students' acceptance of tablet PC and their acceptance had a positive effect on their intention to use tablet PC. Similarly, Cheng, Lam, and Yeung [14] studied acceptance of the Internet using UTAUT and discovered a positive relationship between the perceived usefulness

and behavioral intention. Furthermore, Lai, Huang, Lu, and Chang [15] attempted to understand the effects of website trust, perceived usefulness, and perceived ease-of-use on consumers' intention to make online hotel reservation and their research findings indicated that perceived usefulness positively affected consumers' behavioral intention. Within the UTAUT2 context, performance expectancy is a key determinant to explain consumers' behavioral intention. As a result, the following hypothesis was formulated:

H1: Performance expectancy will positively influence consumers' behavioral intention to make online hotel reservation.

### 2.3. Effort Expectancy (EE) and Behavioral Intention (BI)

According to Sun, Lou, Chao, and Wu [16], users were more likely to accept and use a new technology when it not only was user-friendly but also provided an easy-t-use interface and learning guidance. Similarly, Tsao, Shieh, and Jan [17] indicated when customers compared a new technology with what they had previously used, and found that new technology was easier to use and did not require much training, they were more likely to increase their intention to use the new technology. Past studies showed that perceived ease-to-use has been a determinant factor which drove users to use a new technology, and efforts required to use a new technology directly influenced users' acceptance and implementation of the technology [10,18]. In consistent with prior research, Lai, Huang, Lu, and Chang [15] stated that perceived ease-to-use had impacts on online booking. However, while website trust was added to the model for further analysis, no significant difference was found to exist between perceived ease-to-use and online booking. Effort expectancy is considered a prominent factor that has much influence on behavioral intention within the UTAUT2 model. Therefore, the following hypothesis is proposed based on prior research:

H2: Effort expectancy will positively influence consumers' behavioral intention to make online hotel reservation.

### 2.4. Social Influence (SI) and Behavioral Intention (BI)

Venkatesh et al. [10] indicated that social influence is defined as a degree to which an individual perceived others' belief that they should use a new system. For example, a user decides to make online purchases because people around him are doing the same. Moor and Benbasat [19] discovered that when an individual believed a new technology and system would help maintain and boost his status in a group, he was more likely to use such a technology. Consistent with Moor and Benbasat [19], Tsao, Shieh, and Jan [17] pointed out that peer recommendation, support from management and company, and pressure from the employer influence employees' intention to use property management system. Huang, Lai, Chang, Lu, and Lai [20] had similar research findings suggesting that baseball fans were more likely to use Facebook in order to interact with people with similar interests. Based on previous studies that showed positive relationship between social influence and behavioral intention, the researcher wrote the following hypothesis:

H3: Social influence will positively influence consumers' behavioral intention to make online hotel reservation.

### 2.5. Facilitating Conditions (FC), Behavioral Intention (BI), and Use Behavior (UB)

Facilitating conditions refer to the degree to which an individual believes that an organizational and technical infrastructure exists to support the use of the system [10]. Ajzen [21] indicated that individual behavioral intention enhanced when he believed that he was capable of dominating the technology or resources available to him. Al-Khaldi and Wallace [22] served a good example. They discovered that user attitude, experience, and knowledge toward personal computers had influence on whether knowledge workers were likely to use personal computers. Using technology acceptance model as a research fundamental, Shu and Chuang [23] also reported a positive impact of facilitating

conditions on behavioral intention of users using Wikipedia. As a result, the researcher developed the following hypothesis:

H4a: Facilitating conditions will positively influence consumers' behavioral intention to make online hotel reservation.

H4b: Facilitating conditions will positively influence consumers' use behavior to make online hotel reservation.

### 2.6. Hedonic Motivation (HM) and Behavioral Intention (BI)

Consumers tend to maximize their enjoyment when using innovative products and thus they are more likely to dedicate themselves to such an innovation [24]. Brown and Venkatesh [25] stated that fun and enjoyment were two key factors driving people to accept and use a new technology. Thong, Hong, and Tam [26] indicated that hedonic motivation could be manipulated and transformed to perceived enjoyment, which in turn had impacts on consumer acceptance and use of a new technology. In UTAUT2 model, a significant relationship can be found between hedonic motivation and behavioral intention, and based on this assumption, the researcher made the following hypothesis:

H5: Hedonic motivation will positively influence consumers' behavioral intention to make online hotel reservation.

### 2.7. Price Value (PV) and Behavioral Intention (BI)

Much research and social roles mentioned that price can influence behavioral intention [27]. Chan et al. [28] study also indicated that short messaging services (SMS) prevailed because of its low cost. Similarly, online stores, as compared to real shops, operate with lower cost and generate more profits. It is not difficult to assume a significant relationship between price value and behavioral intention. Consequently, the following hypothesis was developed:

H6: Price value will positively influence consumers' behavioral intention to make online hotel reservation.

### 2.8. Habit Behavior (HT), Behavioral Intention (BI), and Use Behavior (UB)

Using UTAUT2 model to empirically test consumers purchasing airline tickets online, Escobar-Rodríguez and Carvajal-Trujillo [29] showed that habit behavior was an important factor that influenced behavioral intention of consumers purchasing airline tickets on the Internet. In order to know whether consumers would buy airline tickets online or not, behavioral intention was a better predictor than habit behavior. Chong and Ngai [30] studied travelers using local social media on their trips and found that travelers' habit behaviors had significant impacts on their behavioral intention and use behavior. Consequently, researcher using UTAUT2 model concluded that a significant relationship can be found between habit behavior and behavioral intention. Based on prior studies, the researcher formulated the following hypotheses:

H7a: Habit behavior will positively influence consumers' behavioral intention to make online hotel reservation.

H7b: Price value will positively influence consumers' use behavior to make online hotel reservation.

### 2.9. Behavioral Intention (BI) and Use Behavior (UB)

Taylor and Todd [31] referred behavioral intention as perceived attitude, and use behavior as actual action. According to their study, user intention would affect how often they use technology. Past research also indicated that behavioral intention is a major determinant of use behavior [32,33]. Raman and Don [34] studied 320 pre-service teachers using learning management software, and using UTAUT2 model they found that behavioral intention was influential in determining use behavior. Also, Escobar-Rodríguez and Carvajal-Trujillo [29] reported behavioral intention as a major determinant of use behavior in predicting whether consumers would purchase airline tickets on the Internet. As a result, the following hypothesis was proposed:

H8: Behavioral intention will positively influence consumers' use behavior to make online hotel reservation.

### 2.10. Moderating Effects of Gender, Age, Experience within UTAUT2

Gefen and Straub [35] found women rated perceived usefulness to be more than men do. Mikkelsen [36] reported women tended to express more computer anxiety than men. Tsao, Shieh, and Jan [17] also indicated that gender seemed to have moderating effects on the influence of performance expectancy on behavioral intention. Venkatesh et al. [10] stated the influence of performance expectancy on behavioral intention was moderated by age and the effect was stronger for younger men. Similarly, Tsao, Shieh, and Jan [17] discovered that the influence of performance expectancy on behavioral intention was moderated by gender. Research showed a significant relationship between performance expectancy and behavioral intention in UTAUT, and based on prior research findings, the following hypothesis was created:

H9: The relationship between performance expectancy and behavioral intention is moderated by gender and age.

Davis, Bagozzi, and Warshaw [37] studied 107 MBA students using new word processing software, and discovered that the influence of effort expectancy on behavioral intention was moderated by experience. Moreover, Gefen and Straub [35] stated that the relationship between effort expectancy and behavioral intention was moderated by gender, and men were reported to rate perceived ease-of-use more than women. Therefore, the following hypothesis was proposed:

H10: The relationship between effort expectancy and behavioral intention is moderated by gender, age, and experience.

Thompson, Haggins, and Howell [38] studied individual experiences on personal computer utilization, and discovered a moderating effect of personal experience on the relationship between social influence and behavioral intention. In addition, Morris and Venkatesh [39] reported significant moderating effects of experience on the relationship between social influence and behavioral intention. In a different study, they also found a significant moderating effect of gender on the relationship between social influence and behavioral intention when studying acceptance of information system by financial organization employees, and research results showed that women had more influence on the relationship than men. Similarly, Venkatesh et al. [10] reported the influence of social influence on behavioral intention was moderated by age, and the influence was more significant for older workers. Thus, the following hypothesis was created:

H11: The relationship between social influence and behavioral intention is moderated by gender, age, and experience.

Venkatesh et al. [12] highlighted the importance of UTAUT2. It is not difficult to understand that consumers have established a consumption pattern long before their actual consumption behavior. However, what attracts consumers to accept and use a new technology now may have to depend on technology design which increases consumer interests. A variety of research showed that age, gender, and experience of consumers have impacts on their hedonic motivation and behavioral intention, which in turn, influence the use behavior. In addition, price value and behavioral intention are affected by age and gender; on the other hand, individual difference in use behavior and behavioral intention are determined by age, gender, and experience.

Binde and Fuksa [40] studied mobile Internet usage in Latvia, Russia and incorporated several constructs in their UTAUT2: performance expectancy, effort expectancy, social influence, facilitating conditions, and price value. They also added technological support and Internet experience as new constructs. They hypothesized technological support and Internet experience would have impacts on behavioral intention, and use behavior might be affected by facilitating conditions, technological support, and Internet experience. They hypothesized age, gender, and experience would have moderating effects on UTAUT2. Their study subjects included 2000 Latvia citizens, and study results showed that mobile Internet usage was affected by performance expectancy, effort expectancy, social

influence, facilitating conditions, price value, technological support, and Internet experience. The impacts of these constructs on behavioral intention and use behavior were moderated by age, gender, and experience. As a result, following hypotheses were developed:

H12a: The impact of facilitating conditions on behavioral intention is moderated by gender, age, and experience.

H12b: The impact of facilitating conditions on use behavior is moderated by age and experience.

H13: The impact of hedonic motivation on behavioral intention is moderated by gender and age

H14: The impact of price value on behavioral intention is moderated by gender, age, and experience.

H15a: The impact of habit behavior on behavioral intention is moderated by gender, age, and experience.

H15b: The impact of habit behavior on use behavior is moderated by gender, age, and experience.

H16: The impact of behavioral intention on use behavior is moderated by experience.

The following figure illustrates the 16 hypotheses postulated above.

The research model consists of 16 hypotheses and is shown in Figure 1.

## 3. Methodology

### 3.1. Data Collection

The instrument selected for this study was a questionnaire. Questionnaires were distributed to research participants who were customers from 17 international hotels located in four metropolitan areas: Taipei City, New Taipei City, Taichung City, and Kaohsiung City. An employee from each hotel was chosen to distribute 40 questionnaires through convenience sampling. Out of 680 distributed questionnaires, 488 valid responses were returned with returning rate of 71.64%, while questionnaires with missing and incomplete data were eliminated. Valid questionnaires were utilized for data analysis.

### 3.2. Survey Instrument

In this study, the survey questionnaire originally developed by Venkatesh et al. [12] was used and some phrases were modified according to the real hotel environment. The questionnaire was divided into two sections: UTAUT2 and demographic description of study subjects. UTAUT2 contained 36 questions and consists of nine constructs: performance expectancy, effort expectancy, social influence, facilitating condition, hedonic motivation, price value, habit behavior, behavioral intention, and use behavior. The 36 items were measured by a five-point Likert scale ranging from "strong disagree = 1" to "strongly agree = 5". The developed questionnaire was pre-tested to make sure of clearness.

### 3.3. Data Analysis

Our hypotheses testing analysis was based on partial least squares structural equation modeling (PLS-SEM) and was conducted using Warp PLS 4.0 developed by Kock [41]. PLS is a statistical method that bears some relation to principal components regression, and is found in many research areas of information technology, especially those using UTAUT and UTAUT2 [10,12,42].

## 4. Results

### 4.1. Descriptive Statistics of Measurement Items

Table 1 presents the descriptive statistics of participants' background variables. Study respondents comprised of 159 (32.6%) males and 329 (67.4%) females. Of 488 subjects, 137 (28.1%) respondents reported themselves as under 30 by age, 236 (48.4%) as 31 to 45, and 115 (23.6%) as 46 and over. In terms of nationality, 359 (73.6%) respondents reported themselves as Taiwanese while 129 (26.4%) were foreigners. As for education level, 65 (13.3%) respondents reported to have completed high school or less, 270 (55.3%) have completed college, and 153 (31.4%) have completed graduate school or higher. Regarding marital status, 270 (55.3%) respondents were married while 218 (44.7%) were single.

respondents were staying in different hotels during the study period, and 173 (35.4%) respondents were staying in hotels in Taipei City, 77 (15.8%) in New Taipei City, 121 (24.8%) in Kaohsiung City, and 117 (24%) in Taichung City.

**Table 1.** Demographic characteristics of the respondents (N = 488).

| Variable | Frequency | % |
|---|---|---|
| **Gender** | | |
| Males | 159 | 32.6 |
| Females | 329 | 67.4 |
| **Age** | | |
| 18–30 | 137 | 28.1 |
| 31–45 | 236 | 48.4 |
| 46+ | 115 | 23.6 |
| **Taiwan citizen** | | |
| Yes | 359 | 73.6 |
| No | 129 | 26.4 |
| **Education** | | |
| High School Graduate | 65 | 13.3 |
| Bachelor's Degree | 270 | 55.3 |
| Graduate Degree | 153 | 31.4 |
| **Married** | | |
| Yes | 270 | 55.3 |
| No | 153 | 31.4 |
| **Location of Residing Hotel** | | |
| Taipei City | 173 | 35.4 |
| New Taipei city | 177 | 35.8 |
| Taichung City | 117 | 24 |
| Kaohsiung City | 121 | 24.8 |
| **Online Purchases Previous Year (times)** | | |
| 2–5 | 268 | 54.1 |
| 6–0 | 105 | 21.5 |
| 11–15 | 38 | 7.8 |
| 16+ | 81 | 16.6 |
| **Hours Spent on Internet Surfing per week** | | |
| <10 | 158 | 32.4 |
| 11–20 | 135 | 27.7 |
| 21–30 | 75 | 15.4 |
| 30+ | 120 | 26.4 |

Of all the respondents, 173 (35.5%) respondents reported to have less experience using online hotel reservation, 156 (32.0%) reported to have some experience, 123 (26.2%) reported to have much experience, and 31 (6.4%) reported to have very much experience. Regarding to average weekly Internet use, including mobile Internet, 158 (32.4%) respondents spent 10 h or less surfing the Internet, 135 (27.7%) respondents spent 11 to 20 h, 75 (15.4%) respondents spent 21–30 h, and 120 (24.6%) spent 31 h or more. As for online purchase experience during previous year, 268 (54.1%) respondents reported to have made online purchases 2 to 5 times, 105 (21.5%) for 6 to 10 times, 38 (7.8%) for 11 to 15 times, and 81(16.6%) for 16 times or more.

*4.2. Structural Equation Modelling (SEM)*

4.2.1. Measurement Model

The reliability and validity of the study instrument were tested using WarpPLS 4.0 developed by Kock [42], which under PLS, provides two measures of item reliability: composite reliability and Cronbach's. The convergent validity and discriminant validity were conducted to test validity of the instrument according to Hulland [43].

The factor loading of all items from PLS measurement model are all greater than 0.70 indicating good indicators. Composite reliability and Cronbach's $\alpha$ values for all scales exceeded the minimum threshold level of 0.70 [44] indicating the reliability of all scales used in the study (Table 2). As for convergent validity, the square root of average variation extract (AVE) of all values exceeded the minimum threshold level of 0.70 [44] indicating the reliability of all scales used in the study (Table 2). Fornell and Larcker's test [44] for discriminant validity revealed relatively high variances extracted for each factor compared to the interscale correlations, which was an indicator of the discriminant validity of the nine constructs (Table 2).

**Table 2.** Reliability, convergent, and discriminant validity of measurement model.

| Construct | (1) | (2) | (3) | (4) | (5) | (6) | (7) | (8) | (9) | CR [b] | $\alpha$ [c] |
|---|---|---|---|---|---|---|---|---|---|---|---|
| (1) PE | 0.84 [a] | | | | | | | | | 0.91 | 0.86 |
| (2) EE | 0.59 | 0.88 [a] | | | | | | | | 0.93 | 0.90 |
| (3) SI | 0.43 | 0.33 | 0.80 [a] | | | | | | | 0.84 | 0.72 |
| (4) FC | 0.49 | 0.51 | 0.45 | 0.76 [a] | | | | | | 0.85 | 0.76 |
| (5) HM | 0.48 | 0.42 | 0.48 | 0.55 | 0.84 [a] | | | | | 0.91 | 0.86 |
| (6) PV | 0.57 | 0.49 | 0.52 | 0.58 | 0.63 | 0.82 [a] | | | | 0.89 | 0.83 |
| (7) HB | 0.38 | 0.42 | 0.35 | 0.54 | 0.46 | 0.47 | 0.79 [a] | | | 0.87 | 0.80 |
| (8) BI | 0.58 | 0.52 | 0.43 | 0.58 | 0.55 | 0.61 | 0.60 | 0.85 [a] | | 0.91 | 0.87 |
| (9) UB | 0.13 | 0.19 | 0.24 | 0.24 | 0.22 | 0.32 | 0.23 | 0.22 | 1 | 1 | 1 |

Note: PE: performance expectancy; EE: effort expectancy; SI: social influence; FC: facilitating condition; HM: hedonic motivation; PV: price value; HB: habit behavior; BI: behavioral intention; US: use behavior. a: square root of AVE (average variance extracted); b: Composite reliability; c: Cronbach's Alpha.

4.2.2. Structural Model

The evaluation of the structural model is used to examine the sixteen hypothesized relationships. The test results are shown in Table 3 and Figure 2. In line with the value and significance of the path coefficients, PE ($\beta = 0.211$), SI ($\beta = 0.075$), FC ($\beta = 0.194$), HM ($\beta = 0.100$), PV ($\beta = 0.255$), and HB ($\beta = 0.288$) appear to have positive impacts on behavioral intention. FC ($\beta = 0.191$) and HB ($\beta = 0.075$) also appear to have positive impacts on use behavior. As for moderating effects, gender plays as moderators among PE and BI ($\beta = -0.08$), SI and BI ($\beta = 0.111$). Age plays as moderators among EE and BI ($\beta = -0.077$), SI and BI ($\beta = 0.106$), HM and BI ($\beta = -0.099$). Experience plays as moderators among SI and BI ($\beta = -0.091$), PV and BI ($\beta = -0.196$), HB and UB ($\beta = -0.069$).

**Table 3.** Path results of structural model.

| Hypotheses | Paths | Path Coefficient | *p*-Value |
|---|---|---|---|
| H1 | PE → BI | 0.211(β1) | $p = 0.00$*** |
| H2 | EE → BI | 0.025 (β2) | $p = 0.27$ |
| H3 | SI → BI | 0.075 (β3) | $p = 0.03$* |
| H4a | FC → BI | 0.194 (β4a) | $p = 0.00$*** |
| H4b | FC → US | 0.191 (β4b) | $p = 0.00$** |
| H5 | HM → BI | 0.100 (β5) | $p = 0.01$* |
| H6 | PV → BI | 0.225 (β6) | $p = 0.00$*** |
| H7 | HB → BI | 0.288 (β7a) | $p = 0.00$*** |
| H7 | HB →US | 0.075 (β7b) | $p = 0.03$* |
| H8 | BI → US | 0.051 (β8) | $p = 0.13$ |
| H9 | GDR × PE → BI | −0.080 (β9) | $p = 0.02$* |
| H9 | AGE × PE → BI | −0.006 (β10) | $p = 0.44$ |
| H10 | GDR × EE→BI | −0.055 (β11) | $p = 0.09$ |
| H10 | AGE × EE → BI | −0.077 (β12) | $p = 0.03$* |
| H10 | EXP × EE → BI | 0.038 (β13) | $p = 0.17$ |
| H11 | GDR × SI → BI | 0.111 (β14) | $p = 0.00$*** |
| H11 | AGE × SI → BI | 0.106 (β15) | $p = 0.00$*** |
| H11 | EXP × SI → BI | −0.091 (β16) | $p = 0.01$* |
| H12a | GDR × FC → BI | 0.046 (β12) | $p = 0.13$ |
| H12a | AGE × FC → BI | −0.026 (β13) | $p = 0.26$ |
| H12a | EXP × FC → BI | 0.067 (β14) | $p = 0.05$ |
| H12b | AGE × FC → UB | −0.046 (β15) | $p = 0.13$ |
| H12b | EXP × FC → UB | −0.066 (β16) | $p = 0.05$ |
| H13 | GDR × HM → BI | −0.023 (β12) | $p = 0.29$ |
| H13 | AGE × HM → BI | −0.099 (β13) | $p = 0.01$* |
| H14 | GDR × PV → BI | 0.047 (β14) | $p = 0.12$ |
| H14 | AGE × PV → BI | −0.051 (β15) | $p = 0.10$ |
| H14 | EXP × PV→BI | −0.196 (β16) | $p = 0.00$*** |
| H15a | GDR × HB → BI | −0.052 (β12) | $p = 0.10$ |
| H15a | AGE × HB → BI | −0.029 (β13) | $p = 0.24$ |
| H15a | EXP × HB→ BI | 0.016(β14) | $p = 0.34$ |
| H15b | GDR × HB→ UB | 0.004(β15) | $p = 0.46$ |
| H15b | AGE × HB→UB | 0.065 (β16) | $p = 0.05$ |
| H15b | EXP × HB→UB | 0.069(β15) | $p = 0.04$* |
| H16 | EXP × BI→UB | 0.072(β15) | $p = 0.05$ |

Note: *$p < 0.05$; ***$p < 0.001$; PE: performance expectancy; EE: effort expectancy; SI: social influence; FC: facilitating condition; HM: hedonic motivation; PV: price value; HB: habit behavior; BI: behavioral intention; US: use behavior; EXP: experience.

### 4.2.3. Coefficient of Determination ($R^2$)

Coefficient of determination, usually denoted as $R^2$, indicates the percentage of the change occurring in the dependent variable that is explained by the change in the independent variables. It assesses how well a model explains and predicts future outcomes. Thus, high $R^2$ produces precise prediction [45]. Hair, Hult, Ringle, and Sarstedt [46] stated that $R^2$ value can be considered weak (0.25), moderate (0.50), and substantial (0.75). As shown in Figure 2, behavioral intention and use behavior are circled and $R^2$ value is presented. The model explained substantial variance of behavior intention ($R^2$ =0.633), indicating that variables of performance expectancy, effort expectancy, social influence, facilitating conditions, hedonic motivation, price value, and habit explained 63.3% of variance in behavioral intention. Additionally, the model explained weak variance of use behavior ($R^2 = 0.136$), indicating that variables of facilitating conditions, habit, and behavioral intention accounted for 13.6% of variance in use behavior.

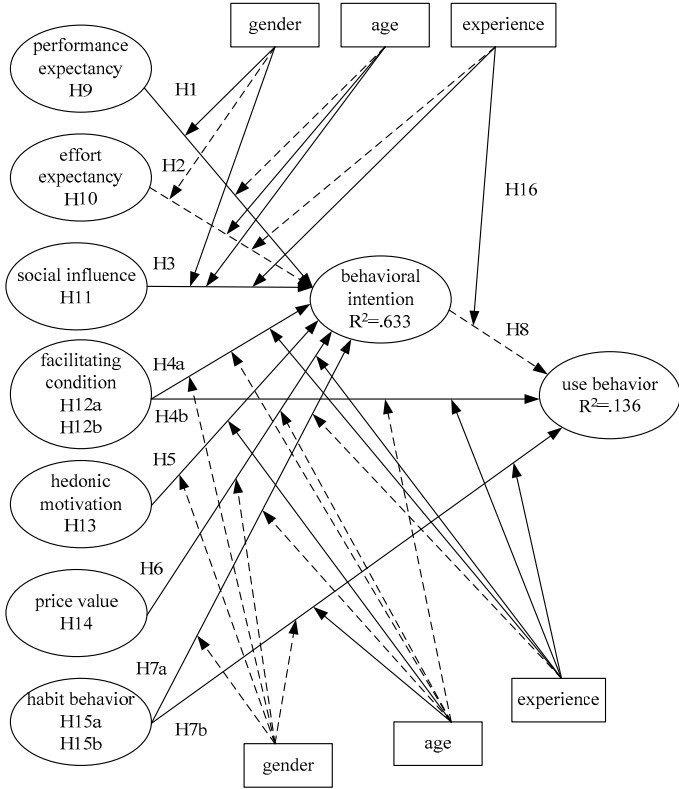

**Figure 2.** Structural equation modelling (SEM) results of the standardized model parameter estimation. (note: "–": path coefficient was not significant; "-": path coefficient was significant).

### 4.2.4. Spurious Correlation Test

In order to explain why hypothesis 2 and hypothesis 8 were not supported by study data, the researcher tested spurious correlation based on Chang [47]. The spurious correlation is also referred to as spurious effect, and the test was intended to explore whether the spurious correlation existed or not. Although hypothesis 2 indicated that effect of effort expectancy on behavioral intention of customers making online hotel reservation was not significant, Figure 3 showed that direct effect of effort expectancy on behavioral intention was significant ($\beta = 0.53$, $p < 0.01$). In addition, as shown in Figure 4 when performance expectancy, social influence, facilitating conditions, hedonic motivation, and price value were respectively included in the model, the results showed significant effect of performance expectancy on behavioral intention. Nonetheless, when habit was added to the model, the effect of effort expectancy on behavioral intention became insignificant ($\beta = 0.05$, $p > 0.05$).

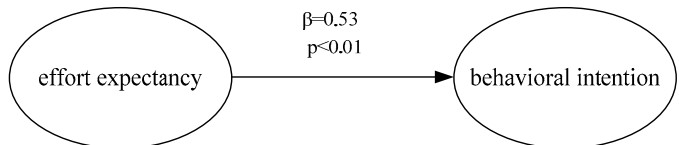

**Figure 3.** Direct effect of effort expectancy on behavioral intention.

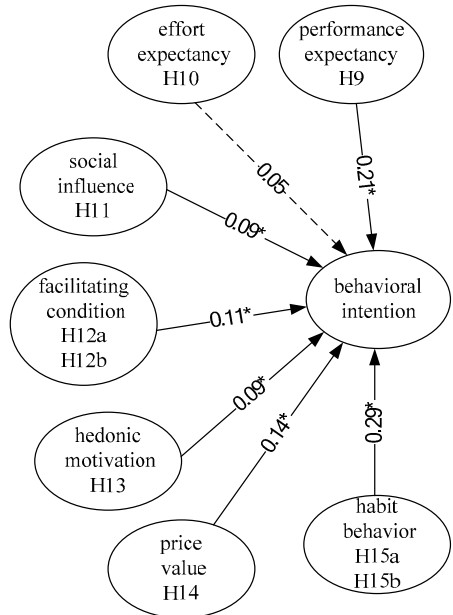

**Figure 4.** Seven independent variables effect on behavioral intention. * $p < 0.05$.

With regard to hypothesis 8 that effect of behavioral intention on use behavior of consumers making online hotel reservation was not found to be significant, Figure 5 reported significant effect of behavioral intention on use behavior ($\beta = 0.30$, $p < 0.01$). As presented in Figure 6 when performance expectancy, effort expectancy, social influence, facilitating conditions, hedonic motivation, price value, and habit were included to the model respectively the study found that effect of behavioral intention on use behavior decreased yet remained significant ($\beta = 0.08$, $p < 0.05$). Experience as a moderator was then included in the model and the study revealed that effect of behavioral intention on use behavior became insignificant ($\beta = 0.06$, $p > 0.05$) (Figure 7). It is concluded that seven variables and experience moderator removed the effect of behavioral intention on use behavior within the model.

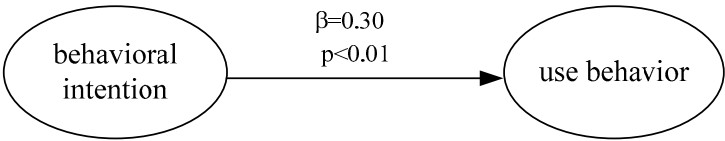

**Figure 5.** Direct effect of behavioral intention on use behavior.

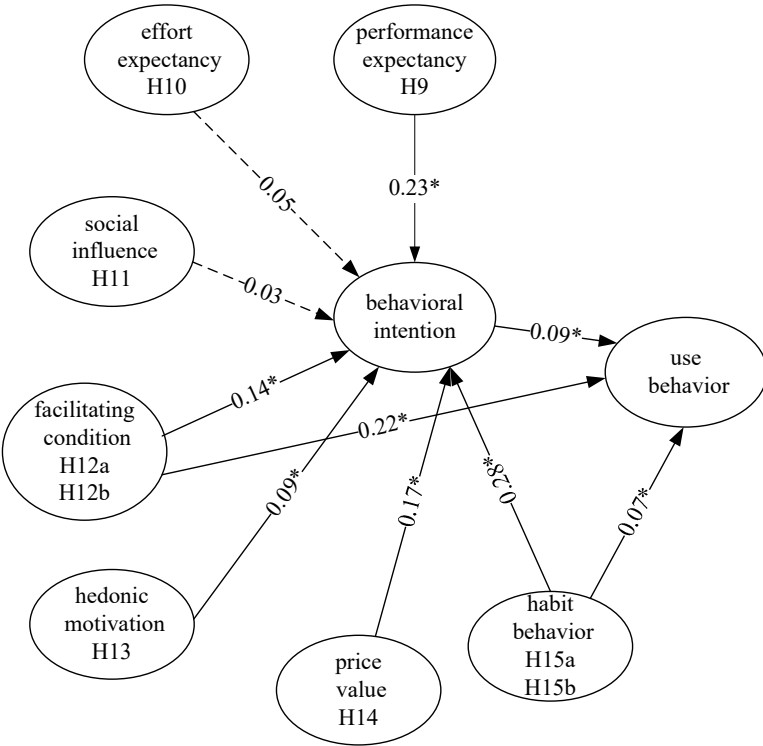

**Figure 6.** Seven independent variables and behavioral intention effect on use behavior. *\*p* < 0.05.

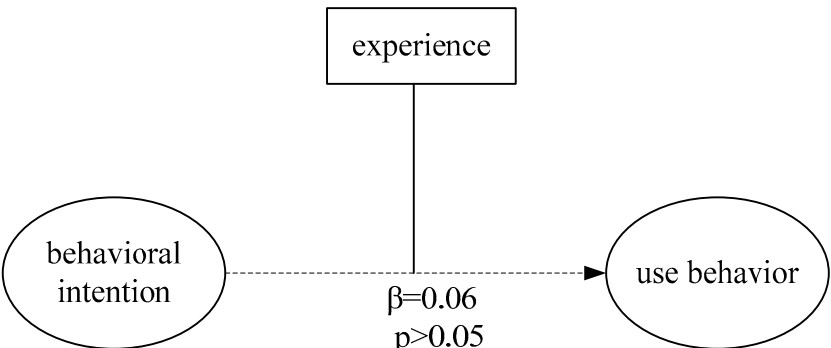

**Figure 7.** Seven variables and experience moderator removed the effect of behavioral intention on use behavior within the model.

## 5. Discussion

### *5.1. Theoretical Contributions*

Research of consumers using new technology and their associated behavior has been widely discussed [7,38,48]. Such research not only helps managers explain and predict user behavior and their acceptance of new technology, but also provides a useful tool for managers to improve technology and further enhance consumer acceptance of technology. Based on Venkatesh, Morris, Davis, and Davis [10] this current study integrated eight technology acceptance models (TAM) and developed UTAUT, a multidimensional research model. Throughout a series of research verification processes, UTAUT has been proved to be more effective than other TAM and it explains up to 70% of variance in the organizational context [10]. Although UTAUT has been employed in a variety of study areas, today it is still insufficient to explain use behavior and behavioral intention in consumers' context. Venkatesh et al. [12] therefore extended UTAUT model and developed a new one, UTAUT2, which is structured based on consumers' technology use and behavioral intention.

Among all the independent variables (i.e., performance expectancy, effort expectancy, social influence, facilitating conditions, hedonic motivation, price value, habit) within the present study, performance expectancy showed no effect on behavioral intention. That means performance expectancy of consumers making online hotel reservation did not have any influence on their behavioral intention. This study result was consistent with Wang, Cheng, and Hsu [49]. Their study further explored whether spurious correlation existed, and found that effect of effort expectancy on behavioral intention within UTAUT model was significant, suggesting that perception of complexity and usability of technology had influence on behavioral intention. Nevertheless, within UTAUT2 model when habit replaces effort expectancy, consumers of international hotels tend to believe they have more access to the Internet and completing online transaction is no longer a problem. As a result, they have better experience using the Internet, and thus effort is not a factor that would have more influence as it used to.

Unlike facilitating conditions and habit that have been reported to have influence on use behavior, behavioral intention has no significant impact on use behavior. Behavioral intention was found to have a significant impact on use behavior when spurious correlation test was included. Contrarily, influence of facilitating conditions and habit on use behavior became insignificant when experience served as a moderator. This is however inconsistent with previous research [29,30,36,40]. In fact Binde and Fuksa [40] was the only research that included moderator within the research model, and more empirical tests can be conducted in the future in order to understand the impact of behavioral intention on use behavior.

Consistent with Escobar-Rodriguez and Carvajal-Trujillo [29] in the context of consumers buying flight tickets online within UTAUT2 model, seven variables have result in different level of influence on behavioral intention. The model explained 63.3% of variance in behavioral intention and 13.6% of variance in use behavior.

When gender as a moderator was included in the research, the relationship between expectancy and behavioral intention was greater for men than for women, and this study finding was consistent with Wu and Lin [50] and Liu [51], suggesting that women are more likely to accept and try hotel online reservation if hotel business is able to help female users quickly identify the rooms they want. Furthermore, consistent with Liu [51], the relationship between social influence and behavioral intention was greater for men than for women. That is, if hotel managers focus more on the idea that hotel online reservation is a new trend, they are more likely to have men make hotel reservation through the Internet.

Increase of age weakens the relationship between effort expectancy and behavioral intention. Older people perceive more the complexity of technology, while younger people perceive more the usefulness of technology. Consequently, an online personal assistant or user guide can be very useful for older people to make hotel online reservation. On the other hand, younger people tend to associate frequent Internet use or any online behavior with a trend or a sense of achievement and happiness and thus are more likely to make online hotel reservation.

Additionally, more experience means weaker relationships between social influence and behavioral intention, between facilitating conditions and use behavior, and between hedonic motivation and behavioral intention. Nonetheless, relationships between facilitating conditions and use behavior, between habit and use behavior, and between behavioral intention and use behavior became stronger when experience accumulated. Consumers with more experience tend not to believe that Internet use should be associated with sense of happiness or achievement. Rather, they have more access to the Internet and believe online behavior is frequent. Therefore they are more likely to make online hotel reservation than those with less Internet experience.

## 5.2. Managerial Implications

Our empirical findings about facilitating conditions and habit have implications for the strategy of consumer IT application vendors. Our results suggest that there are significant impact of facilitating conditions and habit on personal technology use. This suggests that on-going facilitations should be

provided by IT application when consumers need help. For example, through a call center, instant messaging service, or a consumer community can make instant assistance in-need. This surely will enforce their online reservation determination.

As for the significance of the moderated effects in our model suggests that managers can use a market segmentation strategy to facilitate technology. For example, older people rely more on external resources to facilitate their use of the technology. Therefore, a user-friendly help section and a call center provided for senior group can help them to retain and make purchase through the reservation system. Second, we found that younger people are motivated more by the hedonic benefits gained from using technology. This implies that hedonic applications (i.e., music, videos, mobile games) can be bundled together with special promotions to attract younger people.

As for marital status, the study also indicated that married consumers are more interested in using hotel reservation system than those unmarried, and they tend to believe using hotel reservation system can be a pleasant experience. The study also found that married consumers feel the sense of achievement when they are able to find a suitable room type and fair price for the family trip using online hotel reservation system. Therefore, hotel managers can promote a variety of affordable room types to encourage more family trips.

*5.3. Limitations and Future Research*

The first limitation concerns the findings of the test results. Venkatesh et al. [12] expanded the UTAUT and developed UTAUT2 model which incorporated seven independent variables, a moderator, and a dependent variable. Using empirical tests, this study is able to explain consumer behavior in hotel industry. However, spurious correlation is found in some research path and some research findings are inconsistent with previous studies. For example, significant influence of behavioral intention on use behavior became insignificant when facilitating conditions and habit were included and the relationship was moderated by experience. However, as UTAUT2 has not been used much in research and moderators are not usually found in the research, it seems difficult to test spurious correlation. As a result, a more integrated UTAUT2 model should be further tested to better understand consumer acceptance and use of new technology.

Results reported different behavior patterns between domestic and international consumers using online hotel reservation system. Studies with multiple groups within UTAUT2 can be conducted in future to further explore the possible different behavior patterns.

Literature suggests that many consumers make their reservation for domestic hostels via the Internet. It is suggested that future UTAUT2 research can be conducted in a context of hostel consumers to understand their behavior using the hotel reservation.

## 6. Conclusions

The current study showed that in the context of consumers' use of technology of online hotel reservation, the effects of habit, facilitating conditions, and behavioral intention on use behavior are complex. The impact of habit on use behavior is moderated by age and experience. Second, the effect of facilitating conditions on use behavior is moderated by age. Third, the direct effect of behavioral intention on use behavior was insignificant by the moderating effect of experience. Overall, our study confirmed the important roles of habit, facilitating conditions along with the age and experience as moderators influencing online hotel reservation [52].

**Author Contributions:** Conceptualization, C.-M.C. and H.-C.H.; funding acquisition, L.-W.L.; investigation, H.-C.H.; methodology, H.-C.H.; project administration, C.-M.C.; supervision, H.-H.H.; writing—review and editing, H.-H.H.

**Funding:** This research was supported by the Ministry of Science and Technology, Taiwan (MOST 105-2410-H-415 -011).

**Conflicts of Interest:** The authors declare no conflict of interest.

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
