# Peer review of "Factors Influencing Online Hotel Booking: Extending UTAUT2 with Age, Gender, and Experience as Moderators"

_information, doi:10.3390/info10090281_

Round 1
Reviewer 1 Report
The paper covers an interesting topic. It is well-designed and the methodology is adequate.
I have two major concerns about the paper:
(i) the literature review is not enough to provide the adequate background for the several hypotheses evaluated. This section needs a complete revision.
In fact, I believe that in Section 2 the hypotheses put forwarded by the authors could be supported by a stronger theoretical background, namely a stronger link to previous research on the topic.
(ii) The conclusions and implications emerging from the study could also be improved. This part is poor and can be extended.
Regarding the conclusion, I feel that the two aspects mentioned in the first part of Section 5 are insufficient to retain the main conclusions emerging from the study. Since the authors aim to provide some orientations for policy, I think that what is presently discussed is not enough.
Author Response
(i) the literature review is not enough to provide the adequate background for the several hypotheses evaluated. This section needs a complete revision.
In fact, I believe that in Section 2 the hypotheses put forwarded by the authors could be supported by a stronger theoretical background, namely a stronger link to previous research on the topic.
Ans: Thank you for the comments. In order to make the hypotheses clears, we rewrite this section. First, we added a section to introduce the background of UTAUT2, in that section, we also introduced the definition
(ii) The conclusions and implications emerging from the study could also be improved. This part is poor and can be extended.
Regarding the conclusion, I feel that the two aspects mentioned in the first part of Section 5 are insufficient to retain the main conclusions emerging from the study. Since the authors aim to provide some orientations for policy, I think that what is presently discussed is not enough.
Ans: Thank you for the comments. We had rewrote this section significantly and provide more discussion and implications. Please see the discussion and conclusion section for details.
Reviewer 2 Report
The paper presents a questionary-based study related to hotel online booking improvements based on information from Taiwan hotels. Authors propose 16 hypotheses and prove them during the paper.
Unfortunately, the quality and readability of the paper is poor. In the introduction authors should provide the story what the paper about and what novelty it provides for the scientific community, present an example of results utilisation. The main part is written hurriedly, e.g., 3.1. contains only the picture without clear description of it. Results are described in a not convenient way. References mostly are outdated. Related work is missing.
Author Response
Ans: Thank you for comments. We admitted the writing and organization of this paper were not well. At this time, we carefully revised this paper cautiously to present it in good manner. Especially, we carefully proposed the hypotheses according to the UTAUT2 model and even added a section to introduce the model. The results and discussion section were also revised and a conclusion section was added to summarize our results. We also added new references.
Round 2
Reviewer 1 Report
I think that the new version of the paper overcomes my previous concerns.
Therefore, the paper could be accepted for publication.
Reviewer 2 Report
The previous reviewer comments have been fixed.